# α-Synuclein-specific T cell reactivity is associated with preclinical and early Parkinson's disease

Cecilia S. Lindestam Arlehamn [1], Rekha Dhanwani[1], John Pham[1], Rebecca Kuan[1], April Frazier[1], Juliana Rezende Dutra[2], Elizabeth Phillips [3,4], Simon Mallal[3,4], Mario Roederer[5], Karen S. Marder[2], Amy W. Amara [6], David G. Standaert[6], Jennifer G. Goldman[7], Irene Litvan[8], Bjoern Peters[1,9], David Sulzer[10,11,12 ✉] & Alessandro Sette[1,9 ✉]

A diagnosis of motor Parkinson's disease (PD) is preceded by a prolonged premotor phase with accumulating neuronal damage. Here we examined the temporal relation between α-synuclein (α-syn) T cell reactivity and PD. A longitudinal case study revealed that elevated α-syn-specific T cell responses were detected prior to the diagnosis of motor PD, and declined after. The relationship between T cell reactivity and early PD in two independent cohorts showed that α-syn-specific T cell responses were highest shortly after diagnosis of motor PD and then decreased. Additional analysis revealed significant association of α-syn-specific T cell responses with age and lower levodopa equivalent dose. These results confirm the presence of α-syn-reactive T cells in PD and show that they are most abundant immediately after diagnosis of motor PD. These cells may be present years before the diagnosis of motor PD, suggesting avenues of investigation into PD pathogenesis and potential early diagnosis.

[1] Division of Vaccine Discovery, La Jolla Institute for Immunology, La Jolla, CA 92037, USA. [2] Department of Neurology, Columbia University Medical Center, Columbia, NY 10032, USA. [3] Vanderbilt University School of Medicine, Nashville, TN 37235, USA. [4] Institute for Immunology and Infectious Diseases, Murdoch University, Perth, WA 6150, Australia. [5] Vaccine Research Center, National Institute of Allergy and Infectious Diseases, National Institutes of Health, Bethesda, MD 20892, USA. [6] Department of Neurology, University of Alabama at Birmingham, Birmingham, AL 35233, USA. [7] Shirley Ryan AbilityLab, Northwestern University Feinberg School of Medicine, Chicago, IL 60611, USA. [8] Department of Neurology, University of California San Diego, La Jolla, CA 92093, USA. [9] Department of Medicine, University of California San Diego, La Jolla, CA 92093, USA. [10] Department of Neurology, Columbia University, New York, NY 10032, USA. [11] Departments of Psychiatry and Pharmacology, Columbia University, New York, NY 10032, USA. [12] New York State Psychiatric Institute, New York, NY 10032, USA. ✉email: ds43@cumc.columbia.edu; alex@lji.org

Parkinson's disease (PD) is a complex disease with several phases: a preclinical phase that may include physiological or pathological changes, but no symptoms; a prodromal phase that includes non-motor symptoms such as constipation, REM sleep behavior disorder, and hyposmia[1]; and motor PD characterized by the cardinal features of tremor, rigidity, bradykinesia, and postural imbalance. The presence of these motor features allows clinical diagnosis of PD.

The motor symptoms of PD are primarily due to the loss of dopaminergic neurons of the substantia nigra pars reticulata (SNpc)[2]. In postmortem brain, loss of dopaminergic neurons can be detected by stereological analysis and labeling of the dopamine synthetic enzyme tyrosine hydroxylase[3]. A stereological analysis of 20 brains from subjects who died from 1 to 27 years after PD diagnosis indicates that the number of SNpc neurons is decreased by 50–90% during the first four years following diagnosis[3]; while after this period, the number of labeled neurons remains fairly constant. Similarly, the number of neuromelanin-pigmented SNpc cells from the same subjects at 3 to 7 years post diagnosis shows a decrease of 33–80% relative to controls. By 4 years after diagnosis of motor PD, the markers of dopamine axons in the dorsal putamen are virtually absent. These observations are consistent with a premotor phase during which there is extensive damage to SNpc neurons preceding the motor symptoms that typically lead to diagnosis and the identification of clinical disease[1].

The concept of an extensive prodromal phase of PD is further supported by the very common occurrence of symptoms that precede the diagnosis of motor PD, including hyposmia, constipation, mood changes, and REM sleep behavior disorder[1]. Although these symptoms can precede PD diagnosis by decades, their presence is not sufficient to diagnose PD and, although sensitivity is relatively good, the specificity is poor for most subjects[4]. These prodromal non-motor features often associated with PD are thought to be due to the loss of additional catecholaminergic and cholinergic neurons and other pathological changes outside the SNpc[5].

Attempts to develop treatment to slow the progression of PD have so far been unsuccessful. One of the important factors in the lack of success is that it may be difficult to modify the disease when treatment is initiated after the majority of SNpc neurons have already been lost. Thus, identifying effective early predictors of PD is of fundamental importance to developing future therapies[6]. One approach to develop early prediction is to identify additional features associated with the preclinical or prodromal phases of PD, including potential inflammatory manifestations[7,8].

A recent study by our team found that some PD patients possess T cells that recognize specific epitopes derived from the PD-associated protein, α-synuclein (α-syn)[9], indicating the presence of autoimmune features in this disease. Classical autoimmune disorders such as diabetes mellitus type I are well established to display differences in specific T cell types over the course of the disease[10]. Here, we present an analysis of α-syn-specific T cells in two ways: (1) by examining a case study of a single individual with blood samples from many years prior to and during the course of motor PD, and (2) by investigating α-syn-specific T cells in two cohorts of PD patients and age-matched non-PD controls. We detected α-syn-specific T cells in the single case years before clinical diagnosis of motor PD, and a higher abundance of pro-inflammatory T cells in a cross-sectional PD patient population shortly after diagnosis of motor PD. These findings indicate that specific T cell reactivity to α-syn-derived epitopes is a feature of premotor and early motor PD.

## Results

### A case study of α-syn-specific T cell reactivity. We were contacted by an individual who was diagnosed with motor PD in

2009 but was otherwise healthy, and who had for unrelated reasons cryopreserved peripheral blood mononuclear cell (PBMC) samples available that were collected from 1998 to present. The individual had read our previous study[9] and generously donated the previously collected samples for analysis. As in the previous study, PBMCs were stimulated for 14 days in vitro with an α-syn epitope pool (a total of 12 peptides). After 2 weeks, cultures were harvested and stimulated with antigen or PHA, as a control, in a triple-color IFNγ, IL-5, and IL-10 Fluorospot assay. As before, IFNγ was examined as a representative cytokine for CD4+ Th1 cells and CD8+ T cells, while IL-5/IL-4 (in flow cytometry assays) was examined to indicate CD4+ Th2 T cells, and, in addition, IL-10 was analyzed as representative of potential regulatory cells.

We detected α-syn-specific T cell reactivity that was deconvoluted and mapped to the $\alpha\text{-syn}_{61\text{-}75}$EQVTNVGGAVVTGVT peptide in samples from 2005, 2006, and 2015 (Fig. 1a and Supplementary Fig. 1). Since we had access to the HLA type of the donor, we were able to predict HLA binding of this particular peptide to each of the expressed alleles. The EQVTNVG-GAVVTGVT peptide was predicted to bind with high affinity (71 nM) to the HLA DRB1*01:01 allele expressed by the donor (HLA type of case study donor: DRB1*01:01/*03:01, DRB3*01:01, DQB1*02:01/*05:01, DQA1*01:01/*05:01; DPB1*04:01/*04:01). The α-syn-specific response was further characterized by ICS assays in samples collected between 1998 and 2015, which revealed that the response was mediated by CD4+ T cells for both IFNγ and IL-4 (Fig. 1b, c). No α-syn-specific IL-10 production was detected in the ICS assay (Fig. 1d).

We then analyzed the evolution of the α-syn-specific and globally stimulated (PHA control) T cell reactivity. Notably, strong T cell reactivity against α-syn was detected between the years 1998 and 2006, more than a decade before the onset of motor PD symptoms and clinical diagnosis. No samples were analyzed for reactivity between the onset of motor PD symptoms and clinical diagnosis. Significant variability in α-syn-specific responses was observed between 2006 and 2008, including in the positive PHA controls, during which the donor experienced shingles (herpes zoster). Interestingly, the α-syn-specific reactivity detected after diagnosis (2012–2018) was significantly lower than between 1998 and 2006 (Fig. 1e). Overall, there was a significant difference in α-syn-specific T cell reactivity comparing pre- and post-PD onset/diagnosis, while no significant difference was observed for the PHA control (Fig. 1f).

Taken together, this case study revealed strong CD4+ T cell responses against the EQVTNVGGAVVTGVT α-syn epitope. The reactivity was detectable 10 years before onset and diagnosis of motor PD, and was associated with a wide variation in response magnitude. Higher responses were detected before onset/diagnosis, leading us to design a broader study of T cell responses over the course of PD.

### Preferential α-syn-specific T cell reactivity in PD. Previously reported α-syn-specific T cell reactivity was determined in a cohort of individuals recruited by the Columbia University Medical Center (CUMC) and the La Jolla Institute for Immunology (LJI). Here, we enrolled additional donors from RUMC, UCSD, and University of Alabama at Birmingham (UAB) (see "Material and methods"). We determined the α-syn-specific T cell reactivity in PD patients and healthy age-matched controls (HC) from these sites. As before, PBMCs were stimulated for 14 days with an α-syn epitope pool derived from the previous study[9], and IFNγ, IL-5, and IL-10 responses against the epitope pool were measured in IFNγ, IL-5, and IL-10 Fluorospot assays after the in vitro culture. If cell numbers allowed, positive pools were

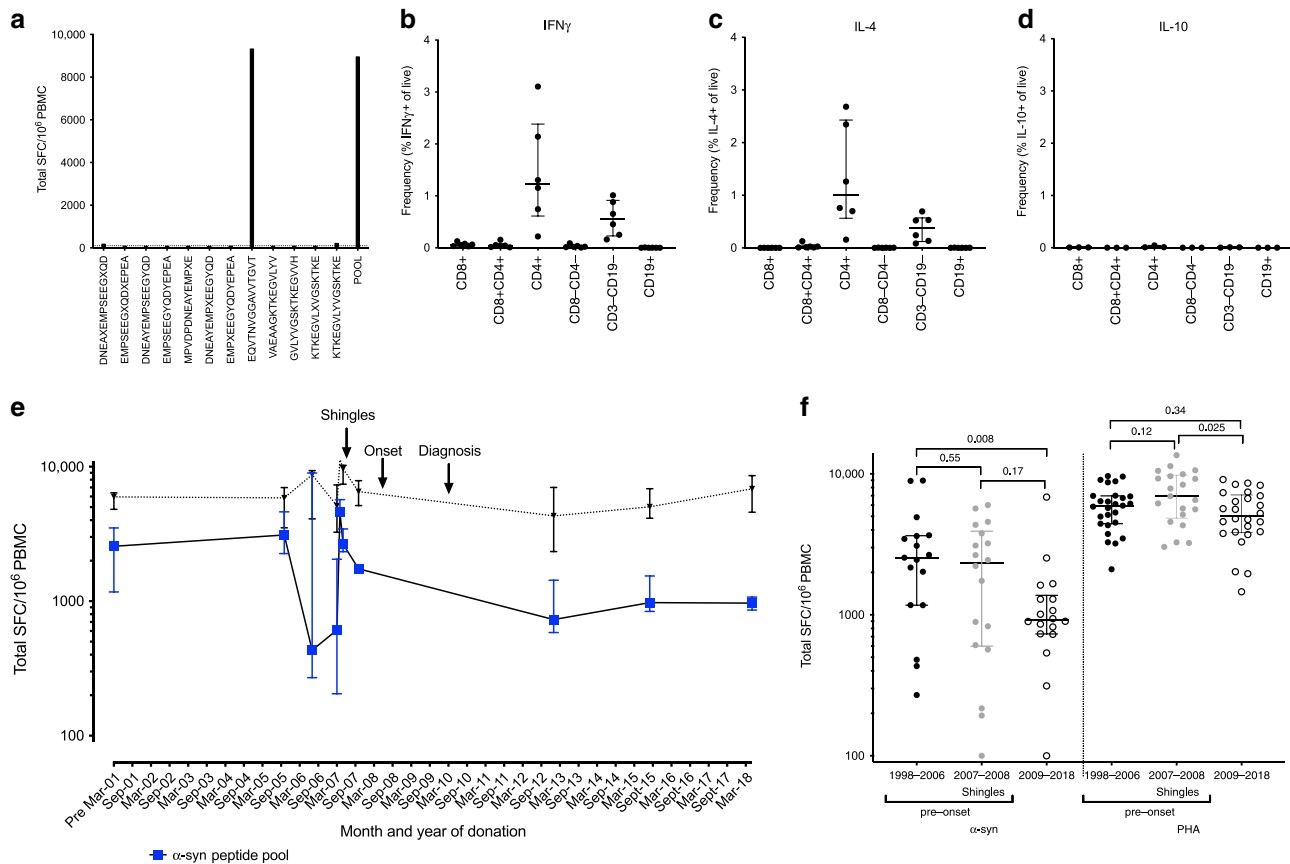

**Fig. 1 α-Syn-specific T cell responses in a longitudinal case study of PD. a** Total magnitude of response expressed as total SFC (sum of IFNγ, IL-5, and IL-10 responses) per $10^6$ cultured PBMC against a pool of α-syn peptides (furthest to the right) and all individual α-syn peptides included in the pool. Representative data from PBMC collected in 2006. **b–d** After eliminating non-lymphocytes and doublet cells by forward and side-scatter, and dead cells by Live/Dead stain, $CD3^+$ T cells were gated based on their CD4 and CD8 expression ($CD8^+$, $CD8^+CD4^+$, $CD4^+$, $CD8^-CD4^-$), and other populations as indicated; $CD3^-CD19^-$ and $CD19^+$ (Supplementary Fig. 5), percentage cytokine of live cells from each of these were plotted. **b** IFNγ ($n = 6$), **c** IL-4 ($n = 6$), and **d** IL-10 ($n = 3$). Each point represents a sample collected between 1998 and 2015 (1998, 2005, 2006, 2007, 2013, and 2015); median ± interquartile range is indicated. **e** Total magnitude of response expressed as total SFC (sum of IFNγ, IL-5, and IL-10 responses) per $10^6$ cultured PBMC against a pool of α-syn peptides or PHA. The time period spans 1998 (pre-2001) to 2018. Median ± interquartile range is indicative of individual experimental replicates. Pre- March-01 $n = 10$, Oct-05 $n = 4$, Jul-06 $n = 3$, Mar-07 $n = 9$, Apr-07 $n = 3$, May-07 $n = 5$, Oct-07 $n = 1$, Jan-13 $n = 4$, Aug-15 $n = 12$, and May-18 $n = 2$. **f** Total magnitude of response divided into pre- (1998–2006) ($n = 17$) and (2007–2008) ($n = 18$), which reflects the period this individual experienced shingles) and post- (2009–2018) ($n = 18$) PD onset/diagnosis for α-syn-specific responses (left) and PHA (right). Each point represents an individual independent experimental replicate; the median ± interquartile range is indicated. Two-tailed Mann–Whitney test. The data are displayed on a logarithmic scale.

deconvoluted to identify the specific peptides that elicited cytokine responses. We previously reported higher α-syn-specific reactivity in PD compared with HC, when comparing the magnitude of response against peptide and participant combinations[9]. The present results in this heterogenous population against the 11 different epitopes deconvoluted after in vitro stimulation confirmed significantly higher magnitude of responses in PD than HC ($p < 0.0001$, Supplementary Fig. 2).

Subsequent analysis focused on responses to the peptide pool with 11 α-syn-derived epitopes; deconvolution of all peptides could not be consistently performed due to limited cell numbers. Comparison of IFNγ, IL-5, and IL-10 production between the two cohorts on a per donor basis revealed no differences when IL-5 and IFNγ were considered separately (Fig. 2a, b), and a significant difference when the IL-10 responses were considered ($p = 0.04$, Fig. 2c). When the overall sum of responses was compared between PD and HC, there were significantly higher overall responses when an arbitrary threshold of 250 SFC was considered ($p = 0.02$; Fisher's exact test, Fig. 2d).

To determine whether the higher α-syn-specific T cell responses in individuals with PD compared with HC occurs in other neurodegenerative diseases, we determined the α-syn-specific T cell reactivity in Alzheimer's disease (AD) patients and HC (Materials and methods). There was no difference in reactivity between these cohorts ($p = 0.15$; two-tailed Mann–Whitney test, and $p = 0.21$; Fisher's exact test with a threshold of 250 SFC, Fig. 2e). We further compared the HC from the PD study to the HC from the AD study, as well as the PD and AD cohorts. No differences were observed by the Fisher's exact test at a threshold of 250 SFC ($p = 0.102$ for HC in PD vs. HC in AD, and $p = 0.196$ for PD vs. AD).

**T cell reactivity is associated with early time points.** To gain further insight into the relation between disease pathogenesis and T cell reactivity, we examined whether the T cell response in a cross-sectional cohort was correlated with time from diagnosis of motor PD. We found that the reactivity was higher closer to PD diagnosis and then waned (Fig. 3a).

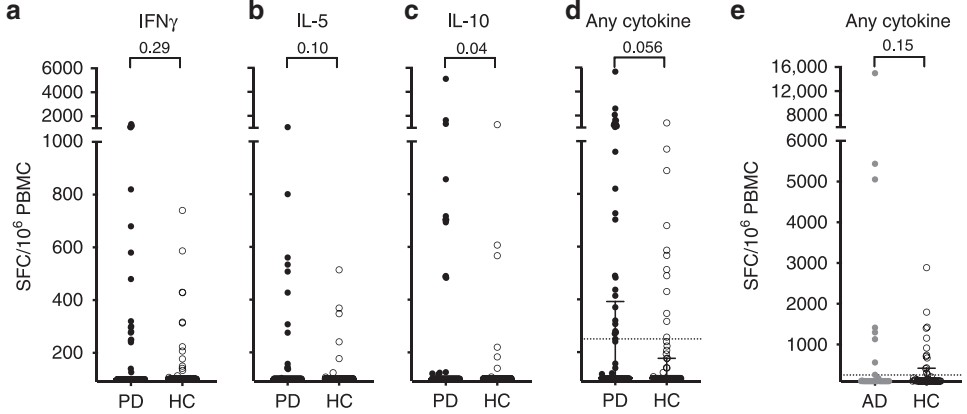

**Fig. 2 Reactivity to α-syn in patients with PD or AD as compared with HC.** Magnitude of responses **a** IFNγ, **b** IL-5, **c** IL-10, and **d** sum of IFNγ, IL-5, and IL-10 against the α-syn peptide pool as SFC per $10^6$ cultured PBMC. Each point represents one participant. Closed circles, patients with PD ($n = 77$); open circles, controls ($n = 69$). One-tailed Mann–Whitney test. As many participants showed no response, numerous points are at the limit of detection (100 SFC). Median ± interquartile range is displayed, but note that median values are at the limit of detection. The top interquartile range is visible in **d**. **e** Sum of IFNγ, IL-5, and IL-10 against the α-syn peptide pool as SFC per $10^6$ cultured PBMC. Each point represents one participant. Gray circles, patients with AD ($n = 38$); open circles, controls ($n = 41$). Two-tailed Mann–Whitney test. As many participants showed no response, numerous points are at the limit of detection (100 SFC). Median ± interquartile range is plotted, but the median values are at the limit of detection. The top interquartile range is visible for HC.

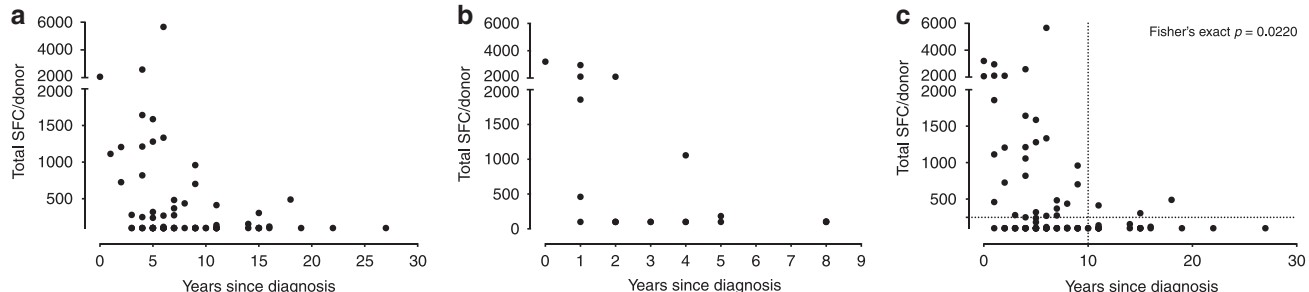

**Fig. 3 Correlation between T cell reactivity and time since diagnosis. a** Cohort 1 ($n = 76$ PD patients from UCSD, RUMC, LJI, and UAB), **b** Cohort 2 ($n = 20$ PD patients from UAB), **c** Cohorts 1 and 2 combined ($n = 96$ PD patients). The Y-axis shows the magnitude of responses (sum of IFNγ, IL-5, and IL-10) against the α-syn peptide pool as SFC per $10^6$ cultured PBMC. The X-axis shows the years since PD diagnosis. Each point represents one donor. Because many participants showed no response, there are overlapping points at the limit of detection (100 SFC). Two-tailed Fisher's exact test.

To confirm the inverse correlation between time from diagnosis and α-syn-specific reactivity, samples derived from a new independent cohort of 20 PD patients, all recruited within 10 years of diagnosis, were analyzed. The results shown in Fig. 3b confirm the previous finding of the reactivity being higher closer to PD diagnosis.

When all the data were considered and plotted together (Fig. 3c), the higher reactivity closer to PD diagnosis was evident. Overall when the frequency of PD patients responding above an arbitrary cut-off of 250 SFC was considered, 39.7% (29/73) of donors diagnosed <10 years ago responded, whereas only 8.6% of donors (3/23) with diagnosis >10 years ago exhibited a response to α-syn ($p = 0.022$, Table 1).

**T cell reactivity is linked to pro-inflammatory cytokines.** Next, we investigated in more detail the relationship between time since diagnosis and IFNγ, IL-5, and IL-10 production. We found that the trend for all cytokines was the same, with the reactivity being higher closer to diagnosis (Fig. 4a–c).

We also calculated the frequency of response using the same cut-offs as above: 250 SFC and 10 years of diagnosis (Table 1). We found that, in the case of IFNγ, 26% (19/73) of patients diagnosed <10 years ago responded as compared with 4.3% (1/23) diagnosed >10 years ago ($p = 0.036$). In the case of IL-5 and IL-10,

**Table 1 Number of responders and nonresponders at 250 SFC and 10-year diagnosis cut-off.**

| Cytokine (250 SFC cut-off) | Time since PD diagnosis | Responders | Nonresponders |
|---|---|---|---|
| Total (sum of IFNγ, IL-5, IL-10) | <10 years | 29 | 44 |
| | ≥10 years | 3 | 20 |
| | *p* value (Fisher's exact) | 0.022 | |
| IFNγ | <10 years | 19 | 54 |
| | ≥10 years | 1 | 22 |
| | *p* value (Fisher's exact) | 0.036 | |
| IL-5 | <10 years | 11 | 62 |
| | ≥10 years | 1 | 22 |
| | *p* value (Fisher's exact) | 0.283 | |
| IL-10 | <10 years | 12 | 61 |
| | ≥10 years | 1 | 22 |
| | *p* value (Fisher's exact) | 0.179 | |

15% (11/73) and 16.4% (12/73) of patients diagnosed <10 years ago responded, respectively, while only 4.3% (1/23) of those diagnosed >10 years ago responded, which corresponded to a nonsignificant trend ($p = 0.283$ and 0.179) in the same direction as observed for IFNγ. Overall, these data indicate a negative (inverse) correlation between the number of years after disease onset and T cell reactivity.

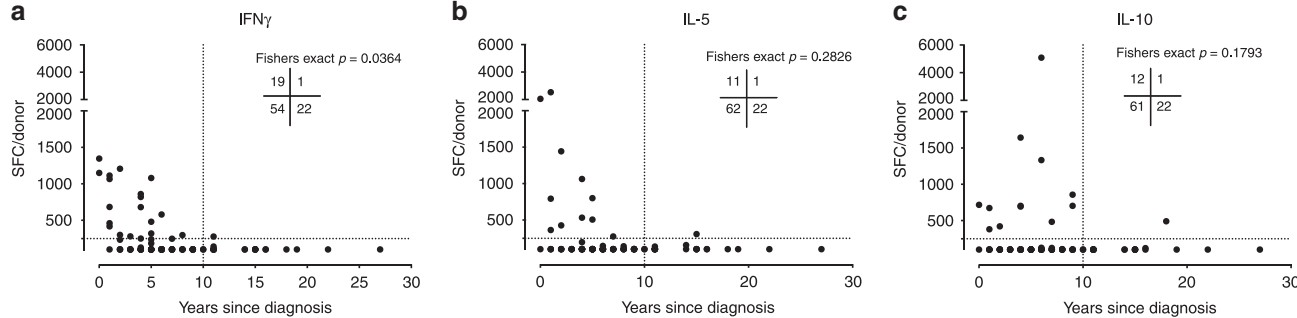

**Fig. 4 Correlation between individual cytokine responses and time since diagnosis.** Cohorts 1 and 2 combined ($n = 96$ PD patients). The $Y$-axis shows the magnitude of responses **a** IFNγ, **b** IL-5, and **c** IL-10, against the α-syn peptide pool as SFC per $10^6$ cultured PBMC. The $X$-axis shows the years since PD diagnosis. Each point represents one donor. Because many participants showed no response, there are overlapping points at the limit of detection (100 SFC). The number of patients in each quadrant is indicated in each graph. Two-tailed Fisher's exact test.

In our previous study[9], although IL-10 production was not investigated in detail, the limited analysis we did indicated that most of the α-syn-specific responses were mediated by CD4+ T cells with one CD8/class I restricted epitope also identified. Here, to address the cell type associated with IL-10 production, 20 PD donors were stimulated in vitro with an α-syn epitope pool for 2 weeks, and then the production of IFNγ, IL-4, and IL-10 along with the lineage-specific markers CD56, CD19, CD14, CD3, CD4, and CD8 was analyzed by flow cytometry. The results indicated that IFNγ and IL-4 were produced primarily by CD4+ T cells, while IL-10 is produced by both CD4+ and CD8+ T cells (Fig. 5a–c), consistent with the previously reported data[9]. Boolean gating of cytokine responses revealed that the majority of responses were from unique T cells producing one of the three cytokines, IFNγ, IL-4, or IL-10 (Fig. 5d), although some cells produced both IFNγ and IL-4.

Because of their potential role in downregulating immune responses, the phenotype of IL-10-producing T cells was examined in more detail with particular attention to whether they expressed markers associated with regulatory T cell (Treg) populations, and conversely, whether Treg populations stimulated with α-syn epitopes secreted IL-10. For this purpose, cells were stained with CD3, CD4, CD8, CD127, and CD25 along with IFNγ, IL-4, and IL-10 after 2-week in vitro stimulation assay. The results demonstrated that IL-10 production was not associated with expression of CD25+CD127lo Tregs, and conversely, that CD25+CD127lo-expressing Tregs do not produce IL-10 in response to the α-syn epitopes (Fig. 5e). Thus, a population of CD25−CD127− T cells was responsible for the α-syn-specific IL-10 response detected.

**LED and age correlate with α-syn-specific T cell reactivity.** We next examined if the α-syn-specific T cell reactivity correlated with clinical and patient-specific characteristics other than time since diagnosis. We evaluated correlations of α-syn-specific T cell reactivity with patient age, sex, and HLA type, as well as with measurements of PD severity such as cognitive function (the Montreal Cognitive Assessment (MoCA)[11]), motor examination (Part III from the Unified Parkinson's Disease Rating Scale (UPDRS)[12]), and medication (levodopa equivalent dose; LED[13]) scores. We found a positive correlation between age and T cell reactivity to α-syn, with patients older than 70 years having more frequent responses ($p = 0.0177$ Fisher's exact test, Fig. 6a). No correlations between age and T cell reactivity to α-syn were found in HC ($p = 0.51$, Supplementary Fig. 3). We also noted a trend toward correlation between T cell responses and sex (Fig. 6b). Only 6 out of 26 (23%) female PD patients tested responded to α-syn, as opposed to 27 out of 71 (38%) male patients, although

this difference was not significant ($p = 0.2279$, with male donors showing a trend toward responding more vigorously than female donors). These results are consistent with an overall increase of autoimmunity with age[14].

Our previous publication found an association with α-syn-specific Y39 T cell responses and HLA DRB1*15:01 and DRB5*01:01[9]. Here, we compared the HLA alleles expressed by PD patients to those expressed by HC to find HLA associations of PD. We found that no HLA allele was associated with PD disease after Bonferroni correction for multiple comparisons (Supplementary Table 1). We also compared the HLA alleles of α-syn responding individuals (either PD or HC with SFC > 250) to nonresponders and found that no significant associations were detected with α-syn responses (Supplementary Table 1).

We next examined potential correlations between T cell reactivity and clinical variables by analyzing the scores for MoCA, which measures cognitive dysfunction; UPDRS, which measures motor and non-motor symptoms and follows the longitudinal progress of PD; and LED, which is designed to quantify anti-Parkinsonian medications. No significant correlation was detected between T cell reactivity and cognitive or motor scores, as MoCA (Fig. 6c) and UPDRS Part III (Fig. 6d) scores were not associated with responses against α-syn. We obtained the complete UPDRS scores (Parts I–IV) available from the cohort recruited at UCSD ($n = 33$) and none of the subscales of UPDRS were associated with responses against α-syn (Supplementary Fig. 4). No association analysis was performed for Hoehn & Yahr stages because 85% of the PD subjects were Stage 2, and only two individuals were rated as Stage 1 and a single subject was rated as Stage 0, or for the Schwab and England Activities of Daily Living scale, because 94% of the PD subjects scored 80–100%. In contrast, our analyses revealed a significant correlation between low LED (<1000 mg/day) and T cell responses against α-syn (Fig. 6e). We reasoned that low LED levels might be simply correlated with early motor disease. Indeed, we found that, as expected, time from diagnosis and low LED were positively correlated (Spearman $r = 0.335$, $p = 0.0016$) (Fig. 6f). We next addressed whether these variables (low LED and time from diagnosis) were independently correlated with α-syn T cell reactivity. For this purpose, we segregated donors as a function of these two variables. We found that responses were essentially limited to donors that had both recent diagnosis and low LED (Fig. 6g). A total of 25/28 responders (SFC > 250) were diagnosed <10 years ago and had a LED score <1000 mg/day, while 33/58 (SFC < 250) did not fit those criteria ($p = 0.0029$ Fisher's exact test).

After noting the positive correlation between age and α-syn responses, we found that a combination of low LED (<1000 mg/day),

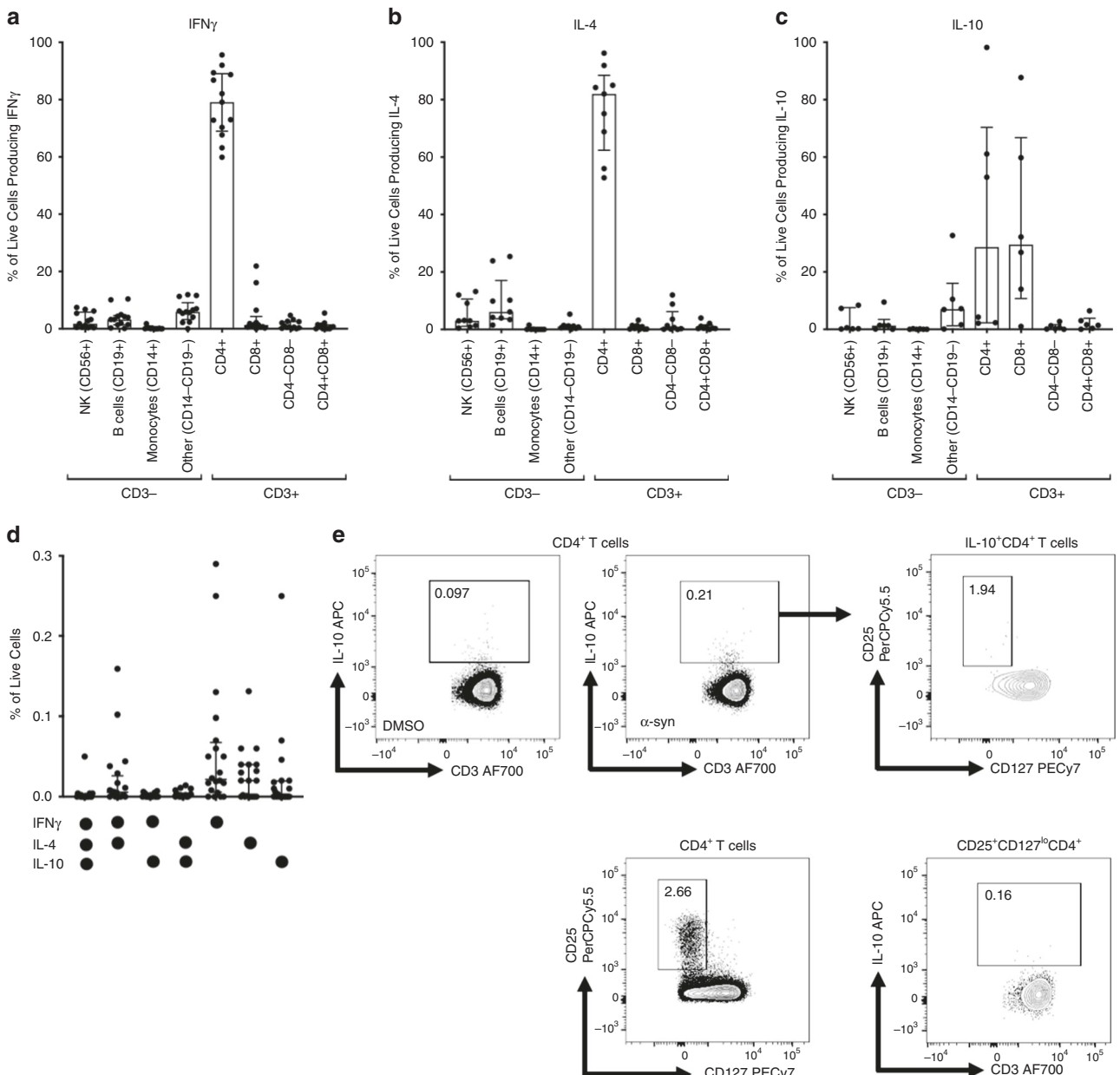

**Fig. 5 IFNγ, IL-4, and IL-10 are produced by distinct T cells in response to α-syn. a–d** After eliminating non-lymphocytes/monocytes and doublet cells by forward and side-scatter and dead cells by Live/Dead stain, cells were gated based on their cytokine expression and then cytokine-expressing cells were gated based on their CD3+ and CD56+ expression. CD3+ T cells were gated based on their CD4 and CD8 expression (CD8+, CD8+CD4+, CD4+, CD8−CD4−), and other populations based on CD3−CD56− and other markers as indicated; CD14−CD19+ (B cells), CD14+CD19− (Monocytes), CD14−CD19− (other) (Supplementary Fig. 5), and percentage live cells expressing cytokine from each of these were plotted. Samples with a frequency of cytokines below 0.02% of live lymphocytes after removal of background were excluded from this analysis. **a** IFNγ (n = 13), **b** IL-4 (n = 9), and **c** IL-10 (n = 6). Median ± interquartile range is indicated. **d** Boolean gating for combinations of cytokines shown as percentage of live cells, n = 20. Median ± interquartile range is indicated. The gating strategy is shown in Supplementary Fig. 5. **e** Plots were gated on IL-10+CD4+ T cells following negative control (DMSO) or α-syn stimulation. The gating strategy leading up to this step is shown in Supplementary Fig. 5. IL-10+ cells were then gated further on CD25 and CD127 expression (top row). Plots were gated on CD25+CD127[lo]CD4+ T cells and then on IL-10 production (bottom row).

time since diagnosis (<10 years), and age (≥70 years) formed an optimum triad of parameters exhibiting maximum sensitivity and specificity. Of the 22 PD donors that meet these three criteria, 15 responded to α-syn, resulting in 68% specificity (Fig. 6h). This particular combination of the three clinical variables: low LED (<1000 mg/day), time since diagnosis (<10 years), and age (≥70 years), captures 15 out of the 28 total PD donors that responded to α-syn, indicating a 54% sensitivity (Fisher's exact p = 0.0001, Fig. 6h).

## Discussion

Here we report that, in patients with motor PD, α-syn-specific T cell responses are highest close to the time of motor PD diagnosis and decline thereafter. These T cell responses are associated with diverse immunological phenotypes, including both inflammatory and potentially regulatory cytokine-secreting cells. We demonstrate in a case study of a single individual in which T cells could be analyzed over a decade that this α-syn-specific inflammatory T cell response was present prior to onset of motor symptoms and

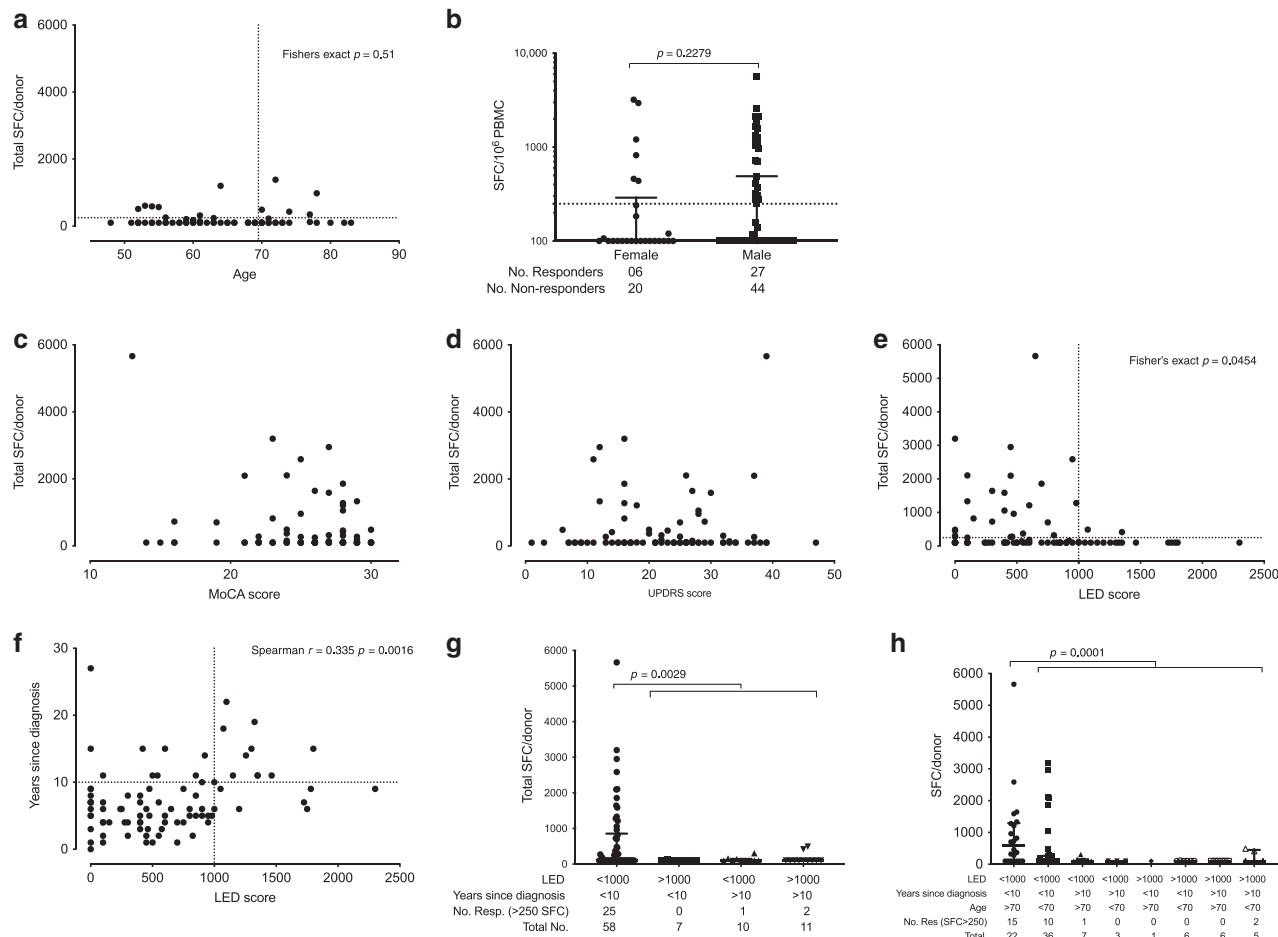

**Fig. 6 Correlation between α-syn-specific T cell reactivity and clinical variables. a** Age ($n = 97$): dotted lines indicate 250 SFC and 70-year-old cut-off for two-tailed Fisher's exact test. **b** Frequency and magnitude of T cell responses in males ($n = 71$) and females ($n = 26$) with PD. The dotted line indicates 250 SFC for Fisher's exact test. **c** MoCA score ($n = 85$), **d** UPDRS score ($n = 85$), and **e** LED score ($n = 86$); dotted lines indicate 250 SFC and 1000 LED cut-off for two-tailed Fisher's exact test. **f** Correlation between LED score and time since diagnosis ($n = 86$); dotted lines indicate 10 years since diagnosis and 1000 LED cut-off for two-tailed Fisher's exact test. Correlation is indicated by Spearman $r$ and associated $p$ value. **g** Segregation of subjects with PD in low (<1000) and high (≥1000) LED score with diagnosis <10 and ≥10 years ago. Number of data points per condition is indicated in the figure (Total No.). Two-tailed Fisher's exact test comparing individuals with <1000 LED, <10 years ago since diagnosis and >250 SFC (left in graph) vs. the three other groups. **h** Segregation of subjects with PD based on LED score (cut-off 1000), years since diagnosis (cut-off 10) and age (cut-off 70). Two-tailed Fisher's exact test comparing individuals with <1000 LED, <10 years ago since diagnosis, >70 years old, and >250 SFC (left in graph) vs. the seven other groups. Total number of PD responding to α-syn peptide pool ($n = 28$). Total SFC is defined as the magnitude of responses (sum of IFNγ, IL-5, and IL-10) against the α-syn peptide pool as SFC per $10^6$ cultured PBMC. **b**, **g**, **h** Median ± interquartile range is indicated.

PD diagnosis. Among the PD cohorts studied after clinical diagnosis of motor PD, we further show that T cell reactivity was positively correlated with age and inversely correlated with LED, but did not correlate with motor and cognitive impairment.

The case study of an individual for which PBMC samples were independently obtained and stored over a 20-year period both before and after clinical diagnosis of motor PD provides a unique opportunity to examine PD disease progression. We found that α-syn-specific T cell reactivity was present before onset of motor symptoms and clinical diagnosis of PD and then declined after. The observation that α-syn reactivity was present during the preclinical phase is compatible with a causal relationship between T cell-mediated inflammation and PD disease. Replication of these findings in larger population-based longitudinal studies will be important, but such cohorts are currently not available. Notably, the case study subject reported having shingles ~1–2 years before PD diagnosis, and we detected a wide variation in α-syn-specific T cell reactivity in samples during that period. It is unclear whether these variations may be due to variations in

sample quality and/or the limited number of cells, which did not allow a repeat of the assay multiple times as for other time points. The variation may also be explained by the shingles episode affecting the α-syn-specific T cell responses. This observation is intriguing in the context of a putative role of viral infections in PD pathogenesis[15,16]. We speculate that the variability in T cell reactivity may have initiated spreading of inflammation and immune responses in the brain that later could not be controlled.

Prompted by these observations, we examined the relation between time from diagnosis of motor PD and α-syn-specific T cell reactivity in two independent cohorts of PD patients and found a significant relationship between time from diagnosis and reactivity. The correlation between α-syn-specific T cell reactivity and time from diagnosis also suggests a pathogenic role of T cell reactivity, but it will be important to examine this further in preclinical and prodromal cohorts, as the mechanism is currently not understood. There is a potential for epitope spreading to occur over time. Due to the limited number of cells, we were unable to identify individual epitopes in all subjects. The limited

data in the case study suggests that there was no epitope spreading, as the epitope specificity remained constant between 2005 and 2015.

The studies in these cohorts also suggested a role for cytokine-producing α-syn-specific T cells in PD progression. It is well known that T cells are very heterogeneous[17], with different subsets functionally distinguished on the basis of differential cytokine secretion. While IFNγ and IL-4/5 secretion are associated with typical inflammatory Th1/Th1* and Th2 subsets[18], other cytokines such as IL-10 that have anti-inflammatory activity are associated with several different subsets, including Tregs, Tr1, and others[19].

In this study, we compared α-syn-specific T cell responses between PD patients and age-matched HC. One limitation to this study is that, while the sample size is relatively large, the PD patients and HC were unbalanced for sex. No difference in α-syn-specific T cell reactivity was found between individuals diagnosed with AD and age-matched HC. Comparison of α-syn-specific T cell reactivity to additional neurodegenerative diseases will reveal whether the reactivity is a marker of neuronal loss and associated neuroinflammation that may occur in multiple neurodegenerative diseases, rather than being specific to PD.

Here, we measured classical inflammatory cytokines as well as IL-10. Significant amounts of IL-10 were produced in addition to IFNγ and IL-5. The IL-10 production suggests the possibility that these cells correspond to a compensatory and regulatory subset. Our limited analysis to date indicates that they are T cells that do not express typical markers of the Treg lineage and are distinct from those producing inflammatory cytokines. These cells might correspond to a specific T cell subset that can be characterized further in future studies. Their isolation and characterization could pave the way for active or passive immunotherapy of PD[20–22].

We also examined the correlation between the presence of different HLA types and PD status (comparing PD with non-PD controls) and in parallel examined the correlation between HLA type and α-syn-specific reactivity in both HC and PD. No significant association was detected when Bonferroni corrections were applied. Thus, the current analysis does not confirm the previous trend (not Bonferroni corrected) for an association with HLA DRB1*15:01 and DRB5*01:01[9]. It should be noted that the previous analysis was performed with a specific DRB1*15:01/DRB5*01:01 restricted epitope, whereas here we studied a pool of 11 α-syn epitopes. In addition, the previous analysis was performed with subjects essentially from a single site, while the present analysis included a more diverse geographical and ethnic basis (New York, Chicago, San Diego, and Birmingham). We also did not find any significant HLA association (Bonferroni corrected) when the individuals from all sites (previous analysis and the current cohorts) were analyzed together. In addition, we did not detect a significant association with the "shared haplotype" recently reported in association with PD[23]. The present study included a total of 164 subjects and was therefore associated with higher statistical power. We conclude that the issue of HLA association with PD should be addressed in further studies with a larger number of subjects analyzed. We are also currently in the process of coupling HLA association studies with TCR repertoire analyses.

Having established a correlation with time since diagnosis, we focused on additional variables of clinical significance. We detected a trend for association of T cell responses and sex, consistent with the bias of disease occurrence in males[24]. We also detected a correlation with reactivity and age, with responses tending to increase with age in the individuals with PD, but not the HC. It is important to emphasize that the α-syn-specific reactivity decreases as a function of time from diagnosis but

increases as a function of age. This observation excludes a general reduction in reactivity due to aging and is more compatible with a general increase in propensity to autoimmunity with increased age as has been previously described[14,25]. Surprisingly, we did not detect significant correlation or association with cognitive function (MoCA) or motor (UPDRS) scores. We propose that this might be reflective of the T cell reactivity largely preceding the clinical stage as in classical autoimmune diseases[26–28], and of T cell reactivity mediating the damage, which does not correlate with disease severity that appears after most of the damage has occurred.

Our analysis detected a strong correlation with low LED (medication scores), and that, while time from diagnosis is as expected correlated with low LED, the two variables also act synergistically. While patients with both low LED and a recent diagnosis exhibit high α-syn-specific reactivity, patients with low LED/distant diagnosis or high LED/recent diagnosis do not show the same high reactivity. This explanation is puzzling, and might relate to high LED/recent diagnosis as a different disease phenotype, with accelerated pathogenesis requiring more medication, or donors where time since diagnosis does not accurately reflect disease onset. One caveat to this study is the risk of potential misdiagnosis of PD in early disease; our PD cohorts were enrolled by fellowship-trained movement disorder specialists and previous studies have shown that the rate of incorrect diagnosis in early PD is around 15%[29,30]. This could also be addressed in a future longitudinal study.

By combining the three variables (age, LED, and time from diagnosis), we achieved the best classification of PD α-syn responders, with almost 70% specificity but low (54%) sensitivity, perhaps because the reactivity of additional antigens in addition to α-syn might be involved in PD pathogenesis. This is consistent with recent reports of reactivity against tau, β-synuclein, and hypocretin[31–33]. The sensitivity could likely be improved by including additional antigens involved in PD pathogenesis, as well as by narrowing the window of time since onset of disease for a diagnostic application. In terms of specificity, it is possible that some of the age-matched controls will eventually develop PD, considering that reactivity in the case study was detected long before diagnosis.

In conclusion, the present study confirms an association of α-syn-specific T cells and PD, and demonstrates that the presence of these T cells is a feature of preclinical and early motor PD. These observations also suggest that the monitoring α-syn-specific T cell reactivity of at-risk populations, such as REM sleep behavior disorder patients and LRRK2 or GBA carriers, might provide a strategy to identify individuals for preventive treatments and immunotherapy approaches. The feasibility of such an approach is reinforced by a recent report that patients undergoing anti-TNF treatments are associated with lower incidence of PD[34].

## Methods

**Study approval**. All participants provided written informed consent for participation in the study. In the case of the AD cohort, all the participants or their authorized representatives provided written informed consent. Ethical approval was obtained from the institutional review boards at La Jolla Institute for Immunology (LJI; Protocol Nos VD-167, VD-124, VD-118, VD-155, and VD-187), Rush University Medical Center (RUMC; Office of Research Affairs No. 16042107-IRB01), University of California San Diego (UCSD; Protocol No. 161224), University of Alabama (UAB; Protocol No. IRB-300001297) and Columbia University Medical Center (CUMC; Protocol No. IRB-AAAQ9714).

**Study subjects' PD and age-matched HC**. We recruited a total of 97 participants with PD and 67 age-matched healthy controls without PD from the greater San Diego (PD, n = 44; 11 LJI, 33 UCSD; HC, n = 35 from UCSD), Chicago (PD, n = 31; HC, n = 32), and Birmingham, Alabama (PD, n = 22, where 20 were used as an independent validation cohort; HC, n = 0) areas; cross-sectional cohort characteristics are shown in Table 2. Blood samples were collected by trained staff.

**Table 2 Demographic characteristics of enrolled PD and matched HC.**

| Characteristics | PD | HC |
|---|---|---|
| Total participants enrolled, n | 97 | 67 |
| Median age (range), years | 67 (45–87) | 64 (48–83) |
| Male, % (n) | 73 (71) | 25 (17) |
| Caucasian, % (n) | 86 (83) | 84 (56) |
| Median age at diagnosis of PD, (range), years | 58 (37–80) | N/A |
| Median years since diagnosis, (range), years | 6 (0–27) | N/A |
| Median UPDRS Part III[a] (range) | 23 (1–47) | N/A |
| Median MoCA[b] (range) | 26 (13–30) | 26 (18–31) |
| Median LED[c] (range) | 520 (0–2300) | N/A |

[a]UPDRS Part III collected at RUMC, UAB, and UCSD.
[b]MoCA collected at RUMC (PD and HC), UAB (PD only), and UCSD (PD only).
[c]LED collected at RUMC, UAB, and UCSD.

**Table 3 Demographic characteristics of enrolled AD and matched HC.**

| Characteristics | AD | HC |
|---|---|---|
| Total participants enrolled, n | 38 | 41 |
| Median age (range), years | 69 (52–89) | 66 (56–92) |
| Male, % (n) | 47 (18) | 46 (19) |
| Caucasian, % (n) | 53 (20) | 90 (37) |
| Median age at diagnosis of AD, (range), years | 63 (47–84) | N/A |
| Median years since diagnosis, (range), years | 4 (0.5–11) | N/A |
| Median MMSE[a] (range) | 22 (16–28) | N/A |
| Median MoCA[b] (range) | 20 (7–26) | 26 (18–31) |

[a]MMSE collected by Precision Med only.
[b]MoCA collected at CUMC only.

The San Diego cohort was recruited by the clinical core at LJI and by the Department of Neurology at UCSD. The Alabama cohort was recruited from the clinical practice of the UAB Movement Disorders Clinic and the Chicago cohort was recruited from the Rush Parkinson's Disease and Movement Disorders Program. PD participants were enrolled on the basis of the following inclusion criteria: moderate to advanced PD; 2 of: rest tremor, rigidity, and/or bradykinesia, PD diagnosis at age 45–75, dopaminergic medication benefit, and ability to provide informed consent, by fellowship-trained movement disorder specialists. Early diagnosis was defined as <10 years prior to sample donation. The exclusion criteria were atypical parkinsonism or other neurological disorders, history of cancer within the past 3 years (not skin), autoimmune disease (except thyroid), and chronic immune-modulatory therapy. Age-matched HC were selected on the basis of age 48–90 and ability to provide informed consent. Exclusion criteria were the same as for PD donors, and, in addition, we excluded self-reported genetic factors (i.e., PD in first-degree blood relative). In the LJI cohort, PD was self-reported. The PD individuals recruited at RUMC, UAB, and UCSD all fulfilled the UK Parkinson's Disease Society Brain Bank criteria for PD. For the PD patients, the median age at diagnosis was 58 years.

The case study experienced onset of symptoms in 2008 and was diagnosed with PD in 2009 when he was 47 years old. He had an episode of shingles in 2007. Upon diagnosis, the case study, a right-hand-dominated man, presented with a year's history of tremors involving the left arm. He also complained of a sore feeling in the arm and some problems while running involving his left leg. There was a family history of essential tremor, but no history of sleeping problems. He denied having constipation or anosmia, and there was no micrographia. Examination revealed a healthy-appearing male who was cognitively intact (mini-mental state examination (MMSE)—30). He had classic intermittent resting tremor involving the left arm, mild to moderate left arm rigidity, and mild slowing of finger tapping on the left. He had diminished left arm swing, but gait was otherwise normal with a UPDRS motor score of 9. He had a Hoehn & Yahr score Stage 1. He started rasagaline 1 month after first the evaluation in 2009. He started Levodopa in December 2015 and had implantation of bilateral subthalamic stimulators in June 2017 due to intractable tremors. Cryopreserved PBMCs derived from either whole blood or leukapheresis donations, isolated using Ficoll-PaqueTM density-gradient centrifugation as below, were provided from various time points between 1998 and 2018.

**Study subjects' AD and age-matched HC.** We recruited a total of 38 participants with AD and 41 age-matched healthy controls without AD from the Alzheimer's Disease Research Center at CUMC (AD, n = 20; HC, n = 21) and Precision Med a Contract Research Organization (AD, n = 18; HC, n = 20). Demographic characteristics of AD and matched HC are listed in Table 3. Blood samples were collected by trained staff.

Subjects from CUMC were diagnosed by neurologists according to the National Institute of Aging and Alzheimer's Association criteria[35]. All AD subjects were probable AD, recruited after at least two clinical visits. Ten AD individuals had neuropsychological testing only; the remaining 10 had neuropsychological testing and positive biomarkers (SPECT scan, FDG PET scan, CSF, or amyloid scan). Twenty-one HC were recruited from the Alzheimer's Disease Research Center at Columbia University. Assessments entail annual neurological and neuropsychological testing (Unified Data Set 3). They were evaluated for at least 2 consecutive years with a normal neuropsychological testing. The neuropsychological testing includes MMSE, MoCA, digit forward and backward, trail making A and B, fluency (phonemic and semantic), multilingual naming test, logical memory (CRAFT), selective reminding test, Benson complex figure (copy and recall), and Rosen, as well as global deficit score and Neuropsychiatric Inventory questionnaire. Five HC were recruited based on a negative amyloid scan and a more limited neuropsychological testing.

The AD cohort recruited by Precision Med were diagnosed according to the National Institute of Neurological and Communicative Disorders and Stroke and the Alzheimer's Disease and Related Disorders Association criteria[36] by a neurologist or internist. In addition, the subjects underwent MRI/CT scans to rule out other causes of cognitive decline. Their MMSE score is ≤28. They exhibit deficits in two or more areas of cognition and have progressive worsening of memory and other cognitive functions including progressive deterioration of specific cognitive functions such as language (aphasia), motor skills (apraxia), and perceptions (agnosia). The diagnosis is further supported by impaired activities of daily living, altered patterns of behavior, and family history of similar disorders. The HC are self-reported without evidence for decline in cognitive functions. The HC have an MMSE of ≥29.

**Peptides.** Peptides were synthesized by A&A, LLC (San Diego, CA) as purified material (>95% by reverse-phase HPLC). Peptides were fifteen 15-mers previously described as T cell epitopes[9], which were combined into two α-syn peptide pools, one for the case study (n = 12) and one for the cross-sectional study (n = 11; Supplementary Table 2). Peptides were aliquoted in small volumes and stored at −20 °C to avoid multiple freeze–thaw cycles.

**PBMC isolation and in vitro expansion.** Venous blood was collected in anticoagulant (e.g., heparin or EDTA)-containing blood bags or tubes. PBMC were purified from whole blood using Ficoll-PaqueTM density-gradient centrifugation, according to the manufacturer's instructions (GE Healthcare Bio-Sciences, Pittsburgh, PA). Cells were cryopreserved in liquid nitrogen suspended in FBS containing 10% (vol/vol) DMSO. For in vitro expansion, cryopreserved PBMCs were thawed in RPMI supplemented with 5% human serum (Gemini Bio-Products, West Sacramento, CA), 1% Glutamax (Gibco, Waltham, MA), 1% penicillin/streptomycin (Omega Scientific, Tarzana, CA), and 50 U/ml Benzonase (Millipore Sigma, Burlington, MA). The cells were then washed and viability was evaluated using the trypan blue dye exclusion method. Briefly, at a density of $2 \times 10^6$ per mL, the cells were plated in each well of a 24-well plate in the presence of a α-syn peptide pool at a concentration of 5 μg/ml and were incubated in a 37 °C humidified $CO_2$ incubator for 2 weeks. Intermittently, every 3 days, cells were supplied with 10 U/ml recombinant human IL-2.

**Fluorospot assay.** After 14 days of culture with α-synuclein peptide pool (5 μg/ml), α-synuclein-specific cellular responses were measured by IFNγ, IL-5, and IL-10 Fluorospot assay with all antibodies and reagents from Mabtech (Nacka Strand, Sweden). Plates were coated overnight at 4 °C with an antibody mixture containing mouse anti-human IFNγ (clone 1-D1K), mouse anti-human IL-5 (clone TRFK5), and mouse anti-human IL-10 (clone 9D7). Briefly, $1 \times 10^5$ cells were added to each well of pre-coated Immobilon-FL PVDF 96-well plates (Mabtech) in the presence of a 5 μg/ml peptide pool and incubated at 37 °C in a humidified $CO_2$ incubator for 20–24 h. Cells from the in vitro culture stimulated with DMSO (corresponding to the percent DMSO in the peptide pool tests) were used to assess nonspecific/background cytokine production and PHA stimulation at 10 μg/ml was used as a positive control. All conditions were tested in triplicate. Fluorospot plates were developed according to the manufacturer's instructions (Mabtech). Briefly, cells were removed and plates were washed six times with 200 μl PBS/0.05% Tween 20 using an automated plate washer. After washing, 100 μl antibody mixture containing IFNγ (7-B6-1-FS-BAM), IL-5 (5A10-WASP), and IL-10 (12G8-biotin) prepared in PBS with 0.1% BSA was added to each well and the plates were incubated for 2 h. The plates were again washed six times with 200 μl PBS/0.05% Tween 20 using an automated plate washer and incubated with diluted fluorophores (anti-BAM-490, anti-WASP-640, and SA-550) for 1 h at room temperature. Finally, the plates were once more washed six times with 200 μl PBS/0.05% Tween 20 using an automated plate washer and incubated with a fluorescence enhancer for 15 min at room temperature. The plates were blotted dry and spots

were counted by computer-assisted image analysis (AID iSpot, Aid Diagnostica GMBH, Strassberg, Germany). Responses were considered positive if the net spot-forming cells (SFC) per $10^6$ PBMC were ≥100, the stimulation index ≥2, and $p ≤ 0.05$ by Student's $t$ test or Poisson distribution test.

**Intracellular cytokine staining (ICS)**. After 14 days of culture, an intracellular cytokine staining (ICS) assay was performed by stimulating PBMCs with 5 μg/ml α-synuclein epitope pool for 2 h in complete RPMI medium at 37 °C. After 2 h, brefeldin A and monensin, each at a concentration of 2.5 μg/ml, were added for an additional 4 h. Cells were incubated for a total of 6 h at 37 °C in humidified $CO_2$ incubator. Stimulated cells were harvested, washed, and stained for cell surface antigens according to the staining panel used (Supplementary Table 3), and fixable viability dye eFluor 506 (eBiosciences, Waltham, MA). After 30 min of staining, cells were washed, fixed using 4% paraformaldehyde and permeabilized using saponin buffer. Cells were stained for cytokines (Supplementary Table 3) in saponin buffer containing 10% FBS. Samples were acquired on a BD LSR II flow cytometer (BD Biosciences, San Jose, CA). Frequencies of $CD3^+$ T cells responding to α-synuclein epitope pool were quantified by determining the total number of gated $CD3^+$ and $cytokine^+$ cells and background values subtracted (as determined from the medium alone control) using FlowJo X Software (FlowJo LLC, Ashland, OR). Combinations of cytokine producing cells were determined using Boolean gating. The gating strategy is described in Supplementary Fig. 5.

**HLA typing**. Participants were HLA typed by an ASHI-accredited laboratory at Murdoch University (Institute for Immunology & Infectious Diseases, Western Australia). HLA typing for class I (HLA A; B; C) and class II (DQA1; DQB1, DRB1 3,4,5; DPB1) was performed using locus-specific PCR amplification of genomic DNA. Primers used for amplification employed patient-specific barcoded primers. Amplified products were quantitated and pooled by subject, and up to 48 subjects were pooled. An indexed (eight indexed MiSeq runs) library was then quantitated using Kappa universal QPCR library quantification kits. Sequencing was performed using an Illumina MiSeq using $2 × 300$ paired-end chemistry. Reads were quality-filtered and passed through a proprietary allele calling algorithm and analysis pipeline using the latest IMGT HLA allele database as a reference. The algorithm was developed by E.P. and S.M. and relies on periodically updated versions of the freely available international immunogenetics information system (http://www.imgt.org) and an ASHI-accredited HLA allele caller software pipeline, IIID HLA Analysis suite (http://www.iiid.com.au/laboratory-testing/). HLA association odds ratios and relative frequencies were calculated using the RATE program[37] available from www.iedb.org.

**Statistical analyses**. Statistical analyses were performed using GraphPad Prism version 7 (GraphPad Software, San Diego, CA). Data were analyzed using non-parametric statistical tests (Spearman test) because the data are not normally distributed. Fisher's exact $t$ test (two-tailed), which provides exact $p$ values for the analysis of contingency tables, was calculated using online GraphPad tool Quick-calcs. Mann–Whitney test (two-tailed) was performed to compare gender-based distribution of PD and frequency of T cell responses in PD and age-matched healthy controls. For all analyses, differences were considered significant if the $p$ value was ≤0.05.

Unless indicated in the figure legend (Fig. 1e, f), each donor sample was included in one experiment.

**Reporting summary**. Further information on research design is available in the Nature Research Reporting Summary linked to this article.

## Data availability
All data generated or analyzed during this study are included in this article and its Supplementary Information. For HLA-typing information from the freely available international immunogenetics information system (http://www.imgt.org), and an ASHI-accredited HLA allele caller software pipeline, IIID HLA Analysis suite (http://www.iiid.com.au/laboratory-testing/) was used.

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

## Acknowledgements

We would like to thank the case study, PD patients, and other participants for donating samples to this study. We would also like to thank Dr. Stephen Grill at the Parkinson's and Movement Disorders Center of Maryland for providing clinical information for the case study. We are also grateful to the La Jolla Institute for Immunology Clinical Studies Group and Flow Cytometry Core. This study was supported by NIH NINDS R01NS095435 (A.S., D.S.), P50NS108675 (D.G.S., A.W.A.), NIH NIA P50AG08702 (ADRC CUMC), the Parkinson's Foundation (A.S., D.S.), the Michael J. Fox (A.S., D.S.), JPB (D.S.) and William F. Richter (D.S.) Foundations and UCSD-LJI Program in Immunology funding (A.S.)

## Author contributions

C.S.L.A., B.P., D.S., and A.S. participated in the design and direction of the study. C.S.L.A., R.D., J.P., and R.K. performed and analyzed experiments. J.R.D., M.R., A.W.A., D.G.S., -K.M., J.G.G., and I.L. recruited participants and performed clinical evaluations. A.F. maintained patient data, records, and assisted in participant recruitment. E.J.P. and S.A.M. coordinated and performed H.L.A. typing. C.S.L.A., R.D., D.S., and A.S. wrote the manuscript. All authors read, edited, and approved the manuscript.

## Competing interests

The authors declare no competing interests.
