## [Peer Review File · Nature Communications]

Reviewers' Comments:

Reviewer #1:

Remarks to the Author:

The results from Arlehamn et al are highly novel and the major claims of the paper justified through several cohorts of subjects, namely α -syn-specific T cell responses are detected early in PD with subsequent decline.

These results should have a major effect on thinking about neuroinflammation in neurological disease. Notably, genetic studies continue to highlight susceptibility genes linked to inflammation, but biologists continue to think about inflammation not in susceptibility to neurodegenerative disease but in progression. This study starts to bridge both worlds, in my opinion, in starting to think about aberrant immune responses as initiators of disease and not just modifiers.

I had only minor comments before in previous submissions that were suggestions to help improve the impact to a wider readership, and these seem to all have been incorporated. I therefore have no other comments.

Reviewer #2:

Remarks to the Author:

In this revised version of the manuscript, the authors could satisfactory work out or clarify several of the points I raised. Being the study conceptually interesting and technically well-performed, I would recommend this work for publication.

Reviewer #3:

Remarks to the Author:

Based on previously published work showing α -syn-specific T cell reactivity in Parkinson's disease, the authors of the current study aimed at exploring a possible temporal relationship between α -syn-specific T cell reactivity and diagnosis of Parkinson's disease. They report on a longitudinal case-study of one patient diagnosed with PD in 2009 and from whom PBMCs were collected serially from 1998 until 2018 and they report 2 cross-sectional cohorts of PD and controls in which α -syn-specific T cell reactivity is studied using a Fluorospot assay. With the two studies, the authors conclude that high α -syn-specific T cell reactivity is present close to the timepoint of diagnosis of Parkinson's disease, declining with disease progression. The results are interesting, but as they are presented at this time a mechanism for the current observation or an application are lacking.

The authors did an adequate effort to study their initial hypothesis. The numbers of patients tested is quite impressive. Even more when taking into account that patient samples are not always easy to collect and process. The clinical information now given is also much clearer. Also, the authors addressed the previous remarks carefully and I believe that it improved the clarity and coherence of the manuscript, but some concerns remain:

Major points:

1. The authors main conclusion from the longitudinal PD case study is that high α -syn-specific T cell reactivity was present well before symptom onset (10 years) and declined after onset (or even before onset, due to the big variation of the data, this is difficult to evaluate) with the authors then mentioning a possible causal relationship (line 348). However, for the cross sectional cohorts the main conclusion is that high α -syn-specific T cell reactivity is present up to 10 years after

diagnosis. This is confusing since for the first part of the manuscript one gets the message that high α -syn-specific T cell reactivity is present in prodromal disease and in the second part the message is that high α -syn-specific T cell reactivity is present up to 10 years after diagnosis particularly in patients >70 years old. Could this reflect different disease phenotypes?

2. I find it difficult to see the application of the current findings for the clinical diagnosis of PD. Mainly due to the fact that also some controls show a high α -syn-specific T cell reactivity (figure 2) and also because the now added AD cohort shows several patients with a much higher α -syn-specific T cell reactivity than PD patients (figure 2 e). I appreciate the fact that the authors did not limit their study to PD and study another neurodegenerative disease, but I think this data also shows that the applicability and mechanism of the findings are not clear. Why is the reactivity of the assay in figure 2e so high compared to the previous experiments?

3. I am confused by the data compared in figure 4 and table 1. As far as I understand, the same data is shown. However, in table 1, IFN γ production is shown to be significantly increased up to 10 years after PD diagnosis whereas only a trend is shown for IL-5 and IL-10 production and no significant difference. This data is not apparent in figure 4. The number of data points shown in figure 4 does not correspond to the same number in Table 1. Is this a problem of data visualization and the overlapping of data points?

4. In figure 6a, what is the rationale for the 70 years of age cutoff? The age parameter is then crucial to define the set of 3 criteria for α -syn responders (>70 years, low LED, <10 years of diagnosis) so it would be important to also show how the non-PD controls compare in α -syn responses according to age. Is α -syn-specific reactivity generally increasing with age?

5. I think a longitudinal study is crucial in this hypothesis. It would define the temporal evolution of high α -syn-specific T cell reactivity in terms of relationship with disease onset (conflicting at this point between the before onset for the longitudinal case, and up to 10 years after diagnosis for the cross sectional cohorts). The results presented by the cross sectional cohorts are interesting but also limited by the fact that the patients are not temporally followed, there is no calculated positive predictive value for the α -syn-specific T cell reactivity and are also limited by the fact that misdiagnosis of PD is higher in patients with early disease, particularly because there is also no correlation with UPDRS scores (and early disease is one of the three variables used by the authors to define the α -syn responders). A longitudinal study would, however, be able to address these issues. How sure can we be that these early disease patients, with low LED have PD?

Minor points:

1. The main conclusion from the longitudinal PD case-study (figure 1) is that high α -syn-specific T cell reactivity was present for 10 years before onset of motor symptoms and decreased after. However, there is high variation in the data, particularly around the shingles episode. In figure 1e, data from before and after onset is presented (in the intervals 98-07 and 08-15) to show a significant difference between pre and post PD. It would be interesting to see also the comparison between 98-06, 06-08 and 08-15 to see the contribution of the episode of shingles for the T cell reactivity.

2. I appreciate the clear clinical description of this longitudinal case.

3. Fig. S2 and fig. 2 legends say median and interquartile range is shown, but this is actually lacking.

4. Line 275-276 the authors argue that α -syn-specific reactivity is higher in males and argue this reflects the sex bias of the diagnosis of PD, which is more frequent in males. I do not understand the rationale for this. Since in this comparison, both male and female patients are already diagnosed with PD according to the same defined criteria, why would the fact that a male is more likely to get diagnosed with PD be relevant?

5. Several times, for eg in line 297 and 426, the authors refer to early PD or early motor PD. In the literature, early PD is a broad term, defined differently by different authors. Some have a cutoff of 2 years since diagnosis, others 7 years since diagnosis; also others refer to early PD as a low H&Y scale. Since this is conflicting in the literature, I would suggest the authors refer specifically to how they define early PD.

Point-by-point reply on NCOMMS-19-34374-T

We thank reviewer 1 and 2 for their positive remarks regarding our manuscript.

Reviewer #3 (Remarks to the Author):

Based on previously published work showing α -syn-specific T cell reactivity in Parkinson's disease, the authors of the current study aimed at exploring a possible temporal relationship between α -syn-specific T cell reactivity and diagnosis of Parkinson's disease. They report on a longitudinal case-study of one patient diagnosed with PD in 2009 and from whom PBMCs were collected serially from 1998 until 2018 and they report 2 cross-sectional cohorts of PD and controls in which α -syn-specific T cell reactivity is studied using a Fluorospot assay. With the two studies, the authors conclude that high α -syn-specific T cell reactivity is present close to the timepoint of diagnosis of Parkinson's disease, declining with disease progression. The results are interesting, but as they are presented at this time a mechanism for the current observation or an application are lacking.

We agree that the mechanism driving the T cell reactivity is currently unknown and have added a statement in the discussion line 366-368: "The correlation between α -syn-specific T cell reactivity and time from diagnosis also suggests a pathogenic role of T cell reactivity, but it will be important to examine this further in preclinical and prodromal cohorts, as the mechanism is currently not understood".

The authors did an adequate effort to study their initial hypothesis. The numbers of patients tested is quite impressive. Even more when taking into account that patient samples are not always easy to collect and process. The clinical information now given is also much clearer. Also, the authors addressed the previous remarks carefully and I believe that it improved the clarity and coherence of the manuscript, but some concerns remain:

We thank the reviewer for these positive remarks.

Major points:

1. The authors main conclusion from the longitudinal PD case study is that high α -syn-specific T cell reactivity was present well before symptom onset (10 years) and declined after onset (or even before onset, due to the big variation of the data, this is difficult to evaluate) with the authors then mentioning a possible causal relationship (line 348). However, for the cross sectional cohorts the main conclusion is that high α -syn-specific T cell reactivity is present up to 10 years after diagnosis. This is confusing since for the first part of the manuscript one gets the message that high α -syn-specific T cell reactivity is present in prodromal disease and in the second part the message is that high α -syn-specific T cell reactivity is present up to 10 years after diagnosis particularly in patients >70 years old. Could this reflect different disease phenotypes?

We apologize for the confusion. It is correct that in the longitudinal donor T cell reactivity is present before symptom onset and declines after onset. In the cross-

sectional study, the reactivity is highest closest to diagnosis and then declines, and we have no way of knowing what the reactivity would be like in the cross-sectional cohort before onset. Therefore, both statements, based on the current results, are true. There is high α -syn-specific T cell reactivity present before onset, which then declines after diagnosis as a function of time.

Ideally our findings would be replicated in larger population-based longitudinal studies and we have been working on this possibility for quite some time. For this to be possible, a new study is required with individuals followed over a long time and the collection of PBMC samples. We have approached the Baltimore longitudinal study of aging lead by the NIA, as well as the Parkinson's progression markers initiative lead by the Michael J Fox Foundation. However, both sample repositories do not have PBMCs of sufficient amounts available for us to study. We have included this information in the discussion line 355-356: "Replication of these findings in larger population-based longitudinal studies will be important, but such cohorts are currently not available".

2. I find it difficult to see the application of the current findings for the clinical diagnosis of PD. Mainly due to the fact that also some controls show a high α -syn-specific T cell reactivity (figure 2) and also because the now added AD cohort shows several patients with a much higher α -syn-specific T cell reactivity than PD patients (figure 2 e). I appreciate the fact that the authors did not limit their study to PD and study another neurodegenerative disease, but I think this data also shows that the applicability and mechanism of the findings are not clear. Why is the reactivity of the assay in figure 2e so high compared to the previous experiments?

We appreciate that the AD cohort shows patients with a much higher α -syn-specific T cell reactivity than PD patients, as the reviewer points out is shown in Fig 2e. This is true for one specific patient (with reactivity above 12,000 SFCs), which appears to be an outlier, out of a total 38 tested. To address whether the reactivity of the assay was different in figure 2e compared to the previous experiments we have performed additional analysis, and added the following statement in the results section line 167-170:

"We further compared the HC from the PD study to the HC from the AD study, as well as the PD and AD cohorts. No differences were observed by two-tailed Mann-Whitney (not shown), or by the Fisher's exact test at a threshold of 250 SFC ($p=0.102$ for HC in PD vs. HC in AD, and $p=0.196$ for PD vs. AD)."

In this study, we did not detect a difference in α -syn-specific T cell reactivity between individuals diagnosed with AD and age-matched HC. As the reviewer points out, the mechanism is not clear, therefore, as stated in the discussion, comparison of α -syn-specific T cell reactivity to additional neurodegenerative diseases will reveal whether the reactivity is a marker of neuronal loss and associated neuroinflammation that may occur in multiple neurodegenerative diseases, rather than being specific to PD. Larger and matched cohorts are needed to study this.

3. I am confused by the data compared in figure 4 and table 1. As far as I understand, the same data is shown. However, in table 1, IFN γ production is shown to be significantly increased up to 10 years after PD diagnosis whereas only a trend is shown for IL-5 and IL-10 production and no significant difference. This data is not apparent in figure 4. The number of data points shown in figure 4 does not correspond to the same

number in Table 1. Is this a problem of data visualization and the overlapping of data points?

Yes, the reviewer is correct, there is a problem with overlapping data points as mentioned in the legend of Figure 4. We have modified figure 4 to include the number of individuals in each quadrant to clarify this further. There are 73 individuals that were diagnosed less than 10 years ago and 23 who were diagnosed more than 10 years ago (as indicated in Table 1). We included Table 1 to allow the reader to see exact numbers even if not all data points are visible in figure 4 as this would allow the reader to see the trend of responses and to visualize the higher reactivity closer to diagnosis.

4. In figure 6a, what is the rationale for the 70 years of age cutoff? The age parameter is then crucial to define the set of 3 criteria for α -syn responders (>70 years, low LED, <10 years of diagnosis) so it would be important to also show how the non-PD controls compare in α -syn responses according to age. Is α -syn-specific reactivity generally increasing with age?

The 70 years of age cut-off is arbitrary based on analysis of the data. As the reviewer suggested, we have performed the same analysis in the HC cohort. We did not find a correlation between age and T cell reactivity to α -syn in the HC. This result is now included in the manuscript as Supplemental figure 3. We state in the results section line 274: “No correlation between age and T cell reactivity to α -syn were found in HC ($p=0.51$)”. Therefore, α -syn-specific reactivity does not generally increase with age. We clarified this in the discussion line 408 stating: “We also detected a correlation with reactivity and age, with responses tending to increase with age in the individuals with PD, but not the HC”.

5. I think a longitudinal study is crucial in this hypothesis. It would define the temporal evolution of high α -syn-specific T cell reactivity in terms of relationship with disease onset (conflicting at this point between the before onset for the longitudinal case, and up to 10 years after diagnosis for the cross sectional cohorts). The results presented by the cross sectional cohorts are interesting but also limited by the fact that the patients are not temporally followed, there is no calculated positive predictive value for the α -syn-specific T cell reactivity and are also limited by the fact that misdiagnosis of PD is higher in patients with early disease, particularly because there is also no correlation with UPDRS scores (and early disease is one of the three variables used by the authors to define the α -syn responders). A longitudinal study would, however, be able to address these issues. How sure can we be that these early disease patients, with low LED have PD?

We agree with the reviewer that longitudinal studies are crucial to investigate this further. As stated above for question 1, these cohorts are not currently available. The reviewer is correct that PD can be misdiagnosed in early disease and this is a caveat of any research enrolling early PD participants. In our cohorts, diagnosis was made by fellowship-trained movement disorder specialists and previous studies have shown that the rate of incorrect diagnosis in most studies of early PD is on the order of 15%. In our cohorts, even if the initial impression was not correct, it is unlikely that there would be a different diagnosis after only a year or two, since this usually takes much

longer. Given the fact that this was not designed as a longitudinal study, we are unable to follow-up with the participants. We have added this caveat to the discussion line 422-425 stating: "One caveat to this study is the risk of potential misdiagnosis of PD in early disease; our PD cohorts were enrolled by fellowship-trained movement disorder specialists and previous studies have shown that the rate of incorrect diagnosis in early PD is around 15% (added references). This could also be able to be addressed in a future longitudinal study".

Minor points:

1. The main conclusion from the longitudinal PD case-study (figure 1) is that high α -syn-specific T cell reactivity was present for 10 years before onset of motor symptoms and decreased after. However, there is high variation in the data, particularly around the shingles episode. In figure 1e, data from before and after onset is presented (in the intervals 98-07 and 08-15) to show a significant difference between pre and post PD. It would be interesting to see also the comparison between 98-06, 06-08 and 08-15 to see the contribution of the episode of shingles for the T cell reactivity.

We thank the reviewer for this suggestion and have replotted the data as suggested. This result is now included in updated figure 1e. As previously stated, there was a wide variation in α -syn-specific T cell reactivity in samples during that period. Importantly, we still see higher α -syn-specific T cell reactivity pre-onset of disease.

2. I appreciate the clear clinical description of this longitudinal case.

Thank you.

3. Fig. S2 and fig. 2 legends say median and interquartile range is shown, but this is actually lacking.

The median and interquartile range was shown, but due to the high number of data points at the limit of detection, the median is also at the limit of detection in fig S2. We have revised the statement in the legend. For figure 2, the top interquartile range is visible in figure d and e. We have revised the legend to better reflect this.

4. Line 275-276 the authors argue that α -syn-specific reactivity is higher in males and argue this reflects the sex bias of the diagnosis of PD, which is more frequent in males. I do not understand the rationale for this. Since in this comparison, both male and female patients are already diagnosed with PD according to the same defined criteria, why would the fact that a male is more likely to get diagnosed with PD be relevant?

We apologize for the lack of clarity. Because males are more frequently diagnosed with PD, we were interested to see whether the magnitude of α -syn-specific T cell reactivity also was higher in males. We have removed the statement of sex bias in incidence of PD.

5. Several times, for eg in line 297 and 426, the authors refer to early PD or early motor PD. In the literature, early PD is a broad term, defined differently by different authors.

Some have a cutoff of 2 years since diagnosis, others 7 years since diagnosis; also others refer to early PD as a low H&Y scale. Since this is conflicting in the literature, I would suggest the authors refer specifically to how they define early PD.

Early motor PD was defined as an individual with 2 of the following: rest tremor, rigidity, bradykinesia and/or dopaminergic medication benefit who was diagnosed less than 10 years ago. Only 3 participants in our study had a H&Y stage of 1 (n=2) or 0 (n=1). These individuals were diagnosed 5, 6, and 7 years ago. We have included this information in the materials and methods, “early diagnosis was defined as less than 10 years ago”, and in the results section line 293-295 “No association analysis was performed for Hoehn & Yahr stages because 85% of the PD subjects were Stage 2, and only two individuals were rated as Stage 1 and a single subject was rated as Stage 0”.

Reviewers' Comments:

Reviewer #3:

Remarks to the Author:

The authors have addressed the previous review remarks accordingly and have made an important effort in clarifying some of the previously raised questions. I am satisfied with the given explanations. This, in my opinion, has improved the clarity and quality of the paper. Particularly, the strengths and caveats of the study are more clearly explained and easier for a reader to understand. Lastly, I would like to thank the authors for their efforts in clarifying the previously raised issues.

REVIEWERS' COMMENTS:

Reviewer #3 (Remarks to the Author):

The authors have addressed the previous review remarks accordingly and have made an important effort in clarifying some of the previously raised questions. I am satisfied with the given explanations. This, in my opinion, has improved the clarity and quality of the paper. Particularly, the strengths and caveats of the study are more clearly explained and easier for a reader to understand. Lastly, I would like to thank the authors for their efforts in clarifying the previously raised issues.

Thank you for the positive remarks.